# Review of Methods for Automatic Plastic Detection in Water Areas Using Satellite Images and Machine Learning

**DOI:** 10.3390/s24165089

**Published:** 2024-08-06

**Authors:** Aleksandr Danilov, Elizaveta Serdiukova

**Affiliations:** Department of Geoecology, Saint Petersburg Mining University, Saint Petersburg 199106, Russia; danilov_as@pers.spmi.ru

**Keywords:** marine pollution, plastic pollution, ocean, Sentinel-2, Earth remote sensing data, waste

## Abstract

Ocean plastic pollution is one of the global environmental problems of our time. “Rubbish islands” formed in the ocean are increasing every year, damaging the marine ecosystem. In order to effectively address this type of pollution, it is necessary to accurately and quickly identify the sources of plastic entering the ocean, identify where it is accumulating, and track the dynamics of waste movement. To this end, remote sensing methods using satellite imagery and aerial photographs from unmanned aerial vehicles are a reliable source of data. Modern machine learning technologies make it possible to automate the detection of floating plastics. This review presents the main projects and research aimed at solving the “plastic” problem. The main data acquisition techniques and the most effective deep learning algorithms are described, various limitations of working with space images are analyzed, and ways to eliminate such shortcomings are proposed.

## 1. Introduction

Water ecosystems play a major role in human life. The functioning of industrial plants, energy, fisheries, water management, development of agricultural production, and other sectors depends on them. Human activity, in turn, has a considerable impact on the state of the aquatic environment. Pollution of water areas, i.e., the introduction of components into the water composition that are not typical of its natural structure, is the consequence of a negative human impact on aquatic ecosystems. One of the most large-scale pollutants of our time is plastic. Rivers are considered the main source of plastic entering the ocean [1]. Plastic can also reach the ocean as a result of fishing and shipping and also due to illegal dumping of waste into the marine environment [2,3].

Water availability and sustainability is one of the seventeen Sustainable Development Goals (SDGs) adopted by the United Nations Member States in 2015. Another goal strives to conserve marine ecosystems [4]. And despite the fact that the likelihood of reaching all the SDGs is poorly assessed in real-world conditions, it is necessary to try to maximize their implementation by 2030 through the development of science and technology [5]. In addition, one of the first marine environmental protection organizations in the world, Ocean Conservancy, says there is a direct link between climate change and the release of plastic waste into the ocean. The reason for this fact is that plastics are produced using fossil fuels, such as oil, gas, and coal. Plastics are already responsible for 3–4% of global greenhouse gas emissions; if they continue to grow, this will triple by 2050 [6].

A second reason for the climate impact of plastic is that small polymer particles destroy bacteria and plankton in the ocean. Such particles are formed as a result of the destruction of large plastics and are called microplastics (MPs) [7]. Due to their size, which is practically indistinguishable to the human eye (<5 mm), MPs are called an “invisible problem”. However, the significance of this trouble has reached global proportions. To date, microplastics are found in all components of the environment, and animals mistakenly feed on them, which leads to the ingestion of this pollutant into the human body [8]. At the same time, the harm of MP particles on health has been repeatedly confirmed by scientists [9,10,11].

There are three ways to fight plastic pollution:Reducing waste entering the ocean from land—improving waste management policies;Identifying debris in the ocean—it allows large accumulations of plastic to be detected to identify the sources and then cleaned up;Cleaning the ocean of debris.

It is necessary to gradually expand the waste management infrastructure worldwide with the average growth of consumer demand for this material, which scientists estimate at 210% until 2060 [1]. The Ocean Conservancy is especially calling for the U.S. to do so, as the world’s first plastic waste-producing country [12].

The crisis situation in this sphere concerns Russia as well. As of 2022, according to a study by the Russian Environmental Operator (REO), the real share of waste utilization in Russia is only 11.9% [13]. At the same time, according to the Federal State Statistics Service, the output of plastic products as of October 2023 compared to the output of plastic products in October 2022 increased by 13.7% [14]. In the Rostov region, for example, there are only three waste processing complexes, so only 5–6% of the total volume of production and consumption of waste in the region is used and neutralized [15]. A comprehensive inventory and analysis of key indicators of the waste management industry was conducted by the REO in 2019–2020, and the following indicators were established as a result:The volume of waste generation is 65 million tons per year;The volume of waste generation per person is 450 kg per year;The volume of solid waste treatment is 18.2 million tons;The volume of solid waste utilization is 2.7 million tons [16].

In the coastal zone of Lake Baikal, waste accumulation has been found both in the most visited places and in hard-to-reach coastal areas, where light and floating garbage is carried by winds, storms, and surf [17]. Therefore, the problem of dumping and accumulation of solid waste, including plastic, in the water area requires a broad solution at both global and regional levels on the part of the state, science, and the public.

However, it is impossible not to mention the positive side of the issue. Dutch inventor Boyan Slat founded an ocean cleanup project called Ocean Cleanup in 2013. The project scales the technology of cleaning plastic from the world’s oceans up to the present with the goal of removing 90% of the debris from the oceans. To achieve this goal, a dual strategy is being used: capturing plastic in rivers to reduce the influx of pollutants and cleaning up the trash already accumulated in the ocean [18].

Another guiding vector in combating the “plastic” problem has been the numerous developments in detecting the accumulation of waste plastic in river systems and the ocean. Modern satellite systems that make Earth remote sensing data (RSD) publicly available allow the identification of pollution.

Since the last century space images have been actively used in various branches of science, including ecology. Industries have the strongest impact on the environment. Remote sensing (RS) methods help to promptly obtain information necessary for environmental monitoring of industrial waste disposal sites, which in the long term allows specialists to assess the negative impact of the mineral resources sector on the environment [19]. In addition, monitoring of oil spills on the water surface is an important area of satellite imagery used in ecology, which is caused by the growth of offshore mining operations [20]. When not balanced ecologically and economically, this type of mining negatively impacts the vulnerable ecosystems of the Arctic, where it has been actively developed recently [21].

Space images have also found a wide application in the study of geological and structural features of hard-to-reach territories in order to identify disturbances and fractured rock zones [22]. In mining, remote sensing is also used to study the dynamics of enterprise operations, check the composition of mining and transportation equipment, and evaluate the results of reclamation of disturbed areas [23]. The impact of large industrial complexes on the environment is usually assessed by analyzing samples, but this method does not allow assessing spatial and dynamic processes, so a combination of methods is used, supplementing the analysis of samples with satellite observations [24] (p. 181). In addition to space data, aerial photography from unmanned aerial vehicles (UAVs) is often carried out, and maps of pollutant distribution are made from the images [25].

Any anthropogenic activity entails certain risks, both to the environment and to human health. In this regard, it is important to assess the probability of occurrence of hazardous events for which space methods are also used. Thus, the processes of bed uplift and subsidence in the territory of the Kirov mine were studied using space radar imaging data. The method used increased the efficiency and reliability of the forecast of geomechanical processes, contributing to the prevention of risks at the enterprise and, consequently, reducing the cost of eliminating negative consequences [26].

These examples show the importance of using satellite data directly during the operation of industrial enterprises and after their liquidation for the assessment of environmental damage. However, it is necessary to emphasize the possibility of using RS data at the initial stage. When planning landscape transformations, i.e., before mining or any other operations, satellite data are used to analyze the ecosystem diversity of the territory [27]. Multitemporal images also help to quantify the forest cover of the region and identify “deforestation” zones, which is important for the sustainable development of the areas under exploitation [28].

In addition to industries, scientists identify urbanization and the growth of cities as one of the significant environmental impact factors. The analysis of urban infrastructure, especially the monitoring of the state of green spaces, is also conducted using RSD [29]. Returning to the problem of plastic pollution in the hydrosphere, it should be noted that cities are the main source of waste generation. Tourism and population growth as a result of urbanization are also adding to them [30].

Satellite-based methods are indispensable in studying the sources of waste discharges into the marine environment and estimating their accumulation. A study showed that the amount of plastic waste is influenced by a number of variables, including demographic factors and economic activity [31]. The physics of plastic transport in the aquatic environment plays an important role in studies of plastic accumulation in water. The dynamics of the distribution of plastic debris depend on two factors:Physical characteristics of plastic (density and size);Environmental characteristics (wind, waves, current).

In one study, the speed of plastic debris movement was estimated to be 6 km/day [32]. Moreover, plastic in the ocean can float on the surface or settle in the water column, which creates significant difficulties with its detection by satellites [33].

In addition to the possibility of covering large areas, an important advantage of using RS methods is the minimization of time. For example, one aerial survey of the Hawaiian Islands required eight analysts to work 688 h for three and a half months to manually interpret all the large macro- and mega-debris [34].

It is also worth noting that modern advanced artificial intelligence (AI) and machine learning (ML) technologies are helping to reduce the time required to process survey results. Deep learning (DL) is the identification of complex patterns by neural networks based on a large dataset. Machine learning is a type of artificial intelligence, which is based on various tools of mathematical statistics, numerical methods, mathematical analysis, etc. Such technologies allow to automatically process a large array of data while increasing the speed and accuracy of the process.

For the task of recognizing plastic in images, the dataset is satellite images or UAV images, which are used by AI or ML to determine the presence of contamination. In deep learning, this is accomplished by having neural networks extract basic features, such as lines, angles, and textures, after which they recognize complex levels of features, such as shapes and boundaries of plastic contamination. Convolutional Neural Networks (CNN) and Recurrent Neural Networks (RNNs) are effective for recognizing objects in images [35]. In machine learning, plastic identification is achieved by training different models to find pixels similar in spectral brightness, i.e., the machine “reads” pixels with plastic on them and finds similar ones by grouping them together. This makes DL and ML a suitable choice for remote sensing data processing.

This approach allows, firstly, to process a large number of images quickly, and secondly, to find features hidden to humans between objects in the images. In addition, neural networks and machine learning involve the creation of models that can be adapted and improved as new data are acquired, which will positively affect the advantage of this approach [36].

Thus, the use of remote sensing and UAV imagery combined with machine learning and artificial intelligence to identify areas of plastic pollution on the water surface could be a key solution to a global problem. In the following, the review describes the technology of plastic identification from satellite images using machine learning based on examples of existing studies. A generalized methodological flowchart showing the sequential steps of this technology is shown in Figure 1.

## 2. State of the Art and Outstanding Methods

### 2.1. Criteria for Choosing a Satellite

Satellite images are the main source of data for the remote method. Satellite data from the Sentinel-2 spacecraft are currently used to identify plastic debris in the images. The spacecraft consists of two identical satellites: Sentinel-2A and Sentinel-2B, developed and operated by the European Space Agency as part of the Copernicus program. Each satellite has a multispectral instrument that measures solar radiation (i.e., it works passively), providing high spatial resolution (10 m, 20 m, or 60 m, depending on the spectral range), which is its main advantage compared to other satellites. Also, the choice of Sentinel is determined by its circulation period: the flyover takes place every five days, which makes it possible to re-explore the necessary territory with a short time interval. It is also important that Sentinel-2 provides free and open data to all users [37]. However, Sentinel satellites do not have coverage over the oceans. Coverage is limited to the near shore water and to inland seas (like the Mediterranean).

It is also known that the Worldview-2 satellite has been successfully utilized. In 2011, after the Japanese earthquake, it was used to monitor the formation of marine plastic debris, but recent studies do not mention the possibility of using Worldview-2 data [32]. The Landsat-8 satellite was also previously used for this purpose, but its spatial resolution is 30 m, which makes it impossible to distinguish small garbage accumulations in the images [38]. In addition, the interval between images of this spacecraft is 16 days. In a given period of time, the object under study may move a significant distance or change in area due to ocean currents. For these reasons, Landsat-8 is not currently used for the purpose of detecting plastics on the surface of water [39]. However, the effectiveness of combining Landsat-8 data with data from the Planet and Sentinel-2 satellites has been proven. The simultaneous use of several satellites, according to the authors, can become an effective tool for monitoring marine plastic debris and can become the basis for a future model for predicting waste accumulation in the ocean [32].

In summary, the choice of the Sentinel-2 satellite for the identification of plastic debris in the marine environment is based on the following facts:High spatial resolution (10 m);Data availability;The orbital period of the satellite is 5 days;The satellite has 13 spectral bands covering visible, near infrared (NIR), and shortwave infrared (SWIR) spectra.

Despite the fact that it is recommended to use Sentinel-2 data, many authors consider that the developed methods for polymer identification are reproducible with other satellites. The only important condition is the identity of the Sentinel-2 spectral bands, as the detection algorithms are based on them [30].

### 2.2. Examples of the Use of Plastic ‘Targets’ for Data Collection and Remote Sensing Experiments on Plastic on Water by Satellites and UAVs

The possibility of detecting plastics in the aquatic environment from unmanned aerial vehicles and the Sentinel-2 satellite was first studied within the framework of the Plastic Litter Project (PLP) in 2018 [40,41]. Also, as part of this study, an experiment was conducted to develop a reference dataset on polymer materials entering the sea.

As an experiment, so-called “targets” consisting of plastic bags, bottles, and natural garbage with built-in GPS sensors were placed on the water in the Gulf of Gera on the island of Lesbos in Greece. The 10 × 10 m targets (imitation of a satellite pixel) were fixed on the water with special anchors. Land control points were placed on the beach. Plastic rafts were released on the water every 5 days for three months while the Sentinel-2 satellite was flying over the area. Together with the satellite data, aerial photography from UAVs was conducted with a 30 min difference (this is not a serious error in the experiment, as noted by the authors) [42]. However, the position of the targets, although firmly fixed, is unstable. Therefore, this time interval should be minimized in future studies.

The experiment was repeated in 2019, but with 1 × 5 m and 5 × 5 m targets. All data from the experiment and information about the project are in the public domain and available online [43]. For 2018 and 2019, it was the only large-scale project of this type. The project continues until the present, with the latest results published for 2023.

Based on existing experience, in 2019, scientists from Greece succeeded in verifying that smaller plastic targets can also be identified from Sentinel-2 images. For this purpose, 3 × 10 m plastic rafts with GPS trackers were installed in the water area of the city of Limassol (Greece) at a distance of 200 m from the shoreline. The study was also conducted using UAV aerial photographs and Sentinel-2 satellite images: the multispectral cameras of the UAV allow for studying the spectral response of plastic, which is then compared with Sentinel-2 spectra [37].

However, in real conditions, plastic accumulations are heterogeneous, with many natural components present. It was possible to obtain a spectrum of mixed material after collecting information about the presence of litter on the coasts of Canada, Vietnam, and Scotland from the scientific literature, and press and social networks [38]. In a recent study, data on plastic pollution was also collected from scientific reports, articles, and social media [39].

Figure 2 shows the difference between the plastic target made for the experiment and plastic pollution in the real environment. As can be seen from the figure, under uncontrolled conditions in the marine environment, plastic accumulates together with natural debris, changes color under the influence of sunlight, and undergoes biofouling, which makes it much more difficult to identify in satellite images.

It should be noted that in the previously mentioned examples, satellite and UAV images are used together. This is due to the fact that there are some limitations and disadvantages in using them separately (Table 1).

Table 2 presents the most significant projects in the field of plastic detection in the water from 2015 to the present.

### 2.3. Features of Satellite Imagery Processing for Floating Plastic Detection

The choice of an effective atmospheric correction algorithm for coastal reservoirs is important for improving the accuracy of detecting floating plastics based on RSD [31]. The ACOLITE atmospheric correction processor, developed by the Royal Belgian Institute of Natural Sciences (RBNSs) for the application of Landsat and Sentinel data in the aquatic environment, performs atmospheric correction and calculates the surface reflection coefficient using water parameters [30]. Atmospheric correction in ACOLITE can be performed using two methods: exponential extrapolation function (EXP) and dark spectrum filtering (DSF). The greatest effectiveness of the latter method has been proven in order to detect plastic in water areas from satellite images [32,39,42].

A “land mask” is used to reduce the probability of pixel misidentification through the calculation of the Normalized Difference Vegetation Index (NDVI) [39]. However, cloud cover can hinder this operation. Modern machine learning methods can improve image quality by reconstructing a high-quality cloud image using various algorithms (Linear Regression—LR, Random Forest Regression—RFR, Support Vector Regression—SVR) based on Sentinel-2 images [51].

If the atmospheric correction is incorrect, a classification error may occur. An example is shown in Figure 3 [44].

### 2.4. Methods and Applications for Obtaining Spectral Characteristics of Various Components of the Marine Environment and Plastics

When applied to remote sensing, spectral analysis means extracting qualitative and quantitative information from the reflectance spectra of a given pixel based on wavelength-dependent properties. In machine learning, spectral characteristics of objects are used to train models, i.e., they are features by which the program determines the object in the image to one or another class. Spectral characteristics of objects are obtained by capturing them with multispectral cameras of unmanned aerial systems (UASs) or with a spectroradiometer. For example, the SVC HR-1024 spectroradiometer was used to obtain spectral characteristics of water surfaces and plastic “targets”. Imaging was performed from heights of 1.5 and 3 m at 20 points in 1 m increments in order to test the spectral response at different heights [37]. In order to obtain spectra of controlled “garbage targets” located on the beach for UAV imaging, the spectrometry session can be carried out in laboratory conditions [52].

The spectral properties of the following materials have been studied by L. Birman [38]:Wood;Seaweed;Sea foam;Plastic;Seawater;Pumice (volcanic rock).

It was found that plastic shows a reflection peak mainly in the near infrared spectrum (NIR), while seaweed reflects light also in the green (560 nm) and red (700–780 nm) bands [38]. The spectral characteristics of plastic bottles at different depths and different types of plastic—polyethylene terephthalate and polyethylene—have been studied in the same way [52]. The spectral reflectance of materials other than plastics allows us to compare the spectral responses of all objects to plastic [39]. Figure 4 shows examples of spectral characteristics of different objects obtained in the studies [32,39,52].

### 2.5. Description of Spectral Indexes Used for the Identification of Floating Plastics

In order to develop a classification model to predict the presence/absence of plastic waste in imagery, a certain set of attributes is required. Such attributes are indexes.

In 2020, B. Burman [38] developed a special index for plastics detection, the Floating Debris Index (FDI), which includes three spectrum bands. The FDI is based on the previously known “Floating algae index” created for the Landsat-8 satellite. Except in this case, the red channel was replaced by RedEdge [36]. The formula for calculation is presented below.
FDI = R_NIR_ − R’_NIR_(1)
R’_NIR_ = R_RE2_ + (R_SWIR1_ − R_RE2_) × ((λ_NIR_ − λ_RED_)/(λ_SWIR1_ − λ_RED_)) × 10(2)
where R_NIR_, R_RE2,_ and R_SWIR1_—denote the reflectance values measured using the satellite per grid corresponding to near infrared (NIR), red edge 2 (RE2,) and shortwave infrared band SWIR-1, respectively, λ_NIR_, λ_RED,_ and λ_SWIR1_ are the wavelengths (in nanometers) corresponding to the NIR, RED, and SWIR-1 bands of the Sentinel-2 satellite presented in Table 3.

In addition to the FDI, the PI—Plastic Index—was developed in 2020. Its use is most appreciated when combined with spectral channels B4 and B8 [37]. Together with it, the authors use the reverse vegetation index RNDVI for the first time, whereas before that only NDVI was known. The NDVI vegetation index makes it possible to distinguish seawater, wood materials, pumice (volcanic rock), and sea foam in the images, but it is not sensitive to the response of plastic. NDVI values range from −1 to 1, with a low value for water and a high value for vegetation. With FDI, plastic objects can be distinguished in the image, but vegetation and other natural materials will create a definition error. Therefore, NDVI and FDI are most often used in combination to achieve the highest efficiency of recognizing materials in the image [38].

The most complete effectiveness of the indexes in the identification of plastics in images is described in the study by M. Duarte. The XGBoost machine learning model was trained on all Sentinel-2 spectral indexes and spectral ranges in order to identify inefficient elements, remove them from the sample, train the model on the remaining components, and, based on this, determine the best combination of channels and indexes. The result was a combination of channels B1 and B8A with indexes NDSI, MNDWI, NDWI, OSI, FDI, WRI, and MARI [39]. Table 4 shows all the indexes and formulas for their calculation, which were used to identify plastics on the surface of the water.

### 2.6. The Most Effective Machine Learning Methods for Detecting Plastic on the Surface of Water

Machine learning (ML) is one of the perspective directions in many fields of science. The data on which the model is trained are images of various objects that may be present in accumulations of floating debris. These include plastic itself, as well as other various objects, including seawater, foam, wood, etc. The attributes on which the model detects general patterns are indexes. The last component of the model is a machine learning algorithm.

B. Basu [30] studied the performance of two unsupervised (K-means and Fuzzy C-means—FCM) and two supervised classification algorithms (Support Vector Regression—SVR and Shape Fuzzy C-Means—SFCM) for identifying floating plastics in coastal water bodies. In order to test the performance of each model, three different sets of attributes were selected. It was found that the performance of each of these algorithms is higher with the largest set of attributes. The most efficient model is SVR [30]. L. Burman used spectral curves to identify macroplastics and the Naive Bayes algorithm to classify mixed materials, which were successfully identified as plastics with 86% accuracy [38].

Support Vector Machine (SVM) and Random Forest (RF) models were also tested to perform classification analysis. The spectral characteristics of different materials and indices were used to develop ML models. For this purpose, a spectral curve profile of marine plastic was created to differentiate floating plastic from other marine debris. Both SVM and RF algorithms performed well in five models and combinations of test cases, but the highest performance was observed for the RF algorithm [53].

The main results in the area of detecting plastic in aquatic environments using machine learning are summarized in Table 5.

Supervised classification algorithms are undoubtedly used more frequently and show higher efficiency, but unsupervised methods are applied when insufficient data are available [30].

Taken together, the results suggest that high-resolution remote sensing imagery and automated ML models can be an effective way to rapidly detect marine floating debris.

## 3. Discussion

After a detailed description of the methods used in the identification of plastics, in this section, we want to draw attention to the drawbacks and limitations of the use of satellites that are highlighted in the studied works.

Cloud cover. The presence of clouds in images is highlighted as a major limitation when working with satellite data in most studies [39,54,55]. Despite the fact that images are usually selected with a filter “cloudiness of <25%”, the obtained data may be insufficient [32]. In addition to clouds themselves, the classification quality can also be affected by cloud shadows, increasing the recognition error of objects in the images. In order to avoid classification errors due to clouds, three machine learning algorithms (LR, RFR, SVR) capable of generating “synthetic” pixels whose spectrum matches that of real objects were tested on Sentinel-2 images. This method can serve as a solution to the problem of image cloudiness. In addition, interference can be caused by sun glare in the images [42]. For more details on atmospheric correction of satellite images see Section 2.3.Limited data availability. A few authors point out the need to expand the existing library of marine debris data [39,51,52]. In this context, “data” refers to images of various kinds of marine debris, on which machine learning models can be trained. Unsupervised classification methods are also known to be used, but their accuracy is lower than that of “learning with a teacher” (see Table 5). With supervised classification, user and producer accuracy is improved, but insufficient data can lead to classification errors. Researchers from different countries are calling for a global spectral database of marine debris data from around the world.Inability to distinguish material type. This is a limitation rather than a disadvantage. If information on plastic types in the contaminated area or for any other qualitative assessment is needed, the data obtained from satellites and UAVs should be supplemented with in situ measurements [56]. The use of remote sensing methods alone, however, can provide a comprehensive picture of the amount of pollution, e.g., to calculate the area or to construct a map of litter density [34,56]. If it is necessary to process images in a large volume, it is advisable to turn to machine learning algorithms in order to automate the process [37].Plastic accumulation. When speaking about the detection of plastic in the water area using space images, we mean its accumulation on the water surface [39]. At small volumes, it is practically impossible to identify it. This is evidenced by the studies described in Section 2.2: the minimum size of a plastic target recognizable on the image is 1 × 5 m. At the same time, the Sentinel-2 pixel coverage should be at least 25% [44]. Taking into account the rapid pace of space industry development, it can be assumed that in the future satellites with lower spatial resolution will be launched, and then, this problem will be solved.Inability to obtain information on submerged litter. The use of satellite imagery has shown a breakthrough in recognizing plastic waste on the water’s surface. However, identifying submerged debris remotely is currently not possible [32]. In the context of solving this problem, it is suggested that efforts should be directed towards the timely removal of marine debris to prevent its submergence due to biofouling and decomposition into microparticles [44].Weather conditions can be an obstacle in the detection of debris. We are not talking here about cloud cover or sun glare but about natural oceanic phenomena, such as storms and strong winds, that can last for a long period of time [32]. This can be a serious problem for the identification of plastics in the high seas, so the focus should be on preventing debris from entering the ocean. For this purpose, it is necessary to introduce a system of monitoring waste accumulations on coastal areas, on beaches, and in river systems. These places are the primary sources of plastic entering the ocean.Classification errors may occur in coastal waters. This is related to the spectral response of water: deep water has a higher reflection coefficient, so plastics are distinguished more effectively and the results are more reliable [39]. In coastal waters, sand and stoniness can interfere with the detection of plastics. However, it should be taken into account that it is technologically easier to conduct a controlled experiment in the coastal area; in addition, storms and strong winds can hinder the detection of plastics at a great distance from the coast (see the previous point).Plastic biofouling. Plastics lose their natural properties when exposed to water for a long time: their structure, shape, and size change, and natural material accumulates on them, which changes the spectral response of plastic [57]. The studies presented in this review prove the effectiveness of solving this problem with the help of machine learning, whose algorithms are capable of automatically decoding images and recognizing suspicious objects. However, no ML model can still be put on the same level as a qualified RS specialist at the moment [58]. ML misses many debris objects, especially if they are biofouled, mistaking them for vegetation, wood, and other natural materials. Further research on improving deep learning models and expanding the database may solve this problem in the future [34].

More than 50 studies were analyzed for the preparation of this review. Figure 5 shows the percentage of all limitations in plastic identification from satellite imagery mentioned in the submitted studies.

Thus, the disadvantages of using satellite optical images provide significant limitations when working with them. In particular, the influence of weather conditions on the survey results. In this regard, it is necessary to pay attention to the possibilities of another type of remote sensing—radar imagery of the Earth. Synthetic Aperture Radar (SAR) uses an active sensor whose detector emits electromagnetic (EM) waves and also records the reflected signal [59]. The EM wave received by the sensor is called the measured backscatter. The SAR image is a two-dimensional visualization of the measured backscatter.

Unlike optical sensors, an SAR sensor can operate both day and night independently of sunlight because it emits the signal itself [60,61]. In addition, electromagnetic waves can penetrate through clouds and ‘see’ under the crown of trees, ensuring operation in any weather conditions. This is probably due to the fact that an SAR sensor uses microwave wavelengths, ranging from K-band (7.5 × 10^−3^ m) to P-band (1 m), while an optical sensor uses wavelengths from visible (4 × 10^−7^ m) to thermal infrared (15 × 10^−6^ m) [62].

However, it should be realized that SAR images cannot be immediately interpreted by the human eye, as they contain only the backscatter signal, and pre-processing of SAR data is a long and complex procedure, including the application of the orbit file, radiometric calibration, gap removal, “multileveling”, speckle correction, and terrain correction [61].

SAR sensing is actively used in various industries, including environmental monitoring: oil spills, urban sprawl, flooding, green space monitoring, etc.

Given the advantages of microwave sensing, the possibility of applying SAR to detect plastic pollution on the water surface is worth considering in future studies.

Future work by the authors. To date, no studies on remote detection of plastic debris on the water surface have been conducted in Russia. In this regard, we would like to note the need to conduct them in Russia. The existing problems with the disposal of solid waste in the country put river and marine ecosystems at high risk of pollution, which, in turn, affects the state of the aquatic environment of neighboring countries [63,64]. The water ecosystems of the Arctic, as the region most exposed to climate change, are particularly vulnerable [21,65,66]. Researchers estimate that the total amount of plastic currently floating in Arctic waters may reach 1200 tonnes [67]. The issue of microplastic pollution is also acute, particularly in the Barents Sea [68]. We want to replicate the experience of our foreign colleagues described in this article in Russia. This may become a significant contribution to the development of a global system for monitoring plastic waste around the world.

We would also like to note that, despite the fact that this article describes mainly the application of machine learning algorithms, deep learning represents a wide range of possibilities for recognizing objects in images. This is due to the fact that neural networks can use pixel data of images and find patterns in them. Therefore, in our future work, we also want to explore the possibility of using artificial intelligence in order to detect plastic from images.

## 4. Conclusions

The use of computer and information technologies has become a necessity in solving most tasks in many branches of science in the modern world. They allow us to simplify and automate the process of working with various data by means of computing and communication.

In the field of Earth remote sensing computers and geoinformation systems, software allows specialists to acquire, process, and interpret space images, which in combination with ground observations, serve as an indispensable tool for solving various tasks. This method is actively used in the field of ecology: space images are used to calculate the areas of mineral resource enterprises’ dumps, to promptly identify forest fires, to determine oil spills, to monitor the condition of tree vegetation, and many others. However, it is necessary to go further and through modern computer technologies to find new ways to solve global environmental problems. Detecting the sources and spread of plastic pollution in the ocean is one of these challenges.

Plastic pollution of the ocean can be called a major environmental disaster of our time. The increasing levels of this material in the marine environment pose serious threats to the marine ecosystem and biodiversity and potentially to humans as well. In the fight against plastic pollution of the planet, ocean clean-up projects are being created, but this is not enough to eliminate the accumulated damage. To date, there are a number of problems associated with the disposal of municipal solid waste, which is the root cause of plastic in water areas.

Accurate detection of plastic litter can help in taking appropriate measures to reduce the amount of plastic in water bodies, providing an opportunity to identify the sources of dumping of solid waste into water bodies, and determining its distribution pathways and locations for elimination of this pollution.

The method is performed by collecting data on the spectral response of materials through the installation of special plastic targets and their imaging from unmanned aerial systems. In the meantime, images of the selected area are acquired from spacecrafts. The European Space Agency Sentinel-2 satellite with a spatial resolution of 10 m is chosen for this purpose. The images are then manually processed by operators and plastic is identified using advanced technologies. These technologies include the new FDI and PI indexes, which “read” the spectral response of the material.

Manual processing of large amounts of data is labor intensive, so the process of detecting debris in images is being automated using machine learning. For this purpose, scientists have tested various algorithms, such as XGBoost, SVR, SVM, and others, the reliability of which shows good results. One of the most high-performance models is Support Vector Regression.

Remote sensing data and advanced machine learning algorithms are effective solutions for identifying large plastic patches, but the potential application of such resources for detecting and recognizing marine floating debris is limited. The main problems when using satellite images are cloudiness and shadows from clouds on images, solar glare, lack of data for training of machine learning models, the possibility to identify plastic only in large volumes (close to the Sentinel-2 pixel size), and others.

In accordance with this, future research is possible along the following directions:Collecting data on the location of plastic pollution around the world, including natural debris, to train IO models;Developing a global database of different types of plastic litter for deep learning;Developing and testing new ML algorithms;Creating a system for monitoring sources of plastic waste entering coastal areas, beaches, and river systems;It is worth considering other satellites in more detail since Sentinel only covers coastal waters and inland seas without surveying the ocean.

A relevant direction in the study of plastic pollution in water areas on the territory of the Russian Federation is the detection of plastic waste in the seas of the Arctic zone.

In the future, the creation of a global system for real-time monitoring of rubbish in the ocean in combination with improved water treatment technologies may become the key to solving the global plastic problem.

## Figures and Tables

**Figure 1 sensors-24-05089-f001:**
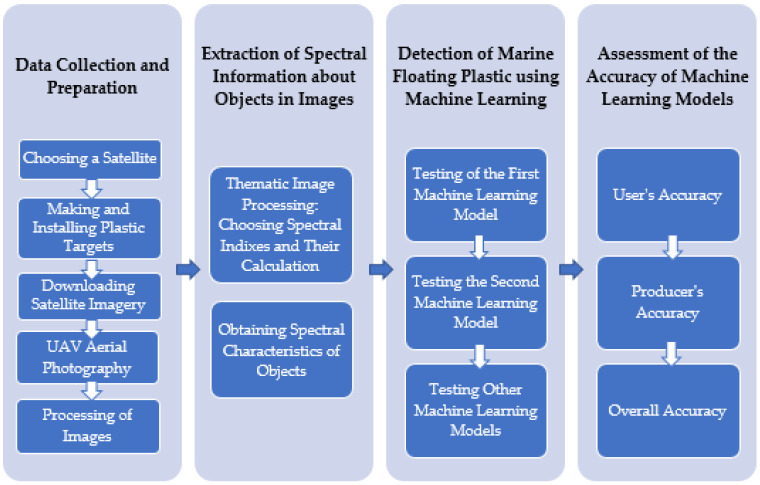
Methodological flowchart showing the sequential steps for automatic detection of plastic waste in the ocean [compiled by the authors].

**Figure 2 sensors-24-05089-f002:**
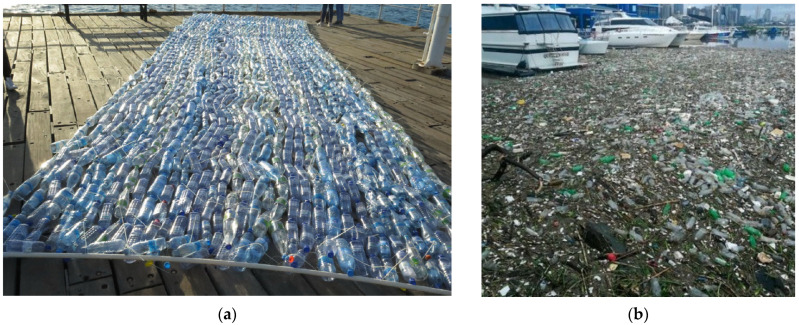
Comparison of plastic accumulations in experimental and real conditions (**a**) Experimental plastic float [37]; (**b**) Pollution of Durban Harbor after the flood (23 April 2019) [38].

**Figure 3 sensors-24-05089-f003:**
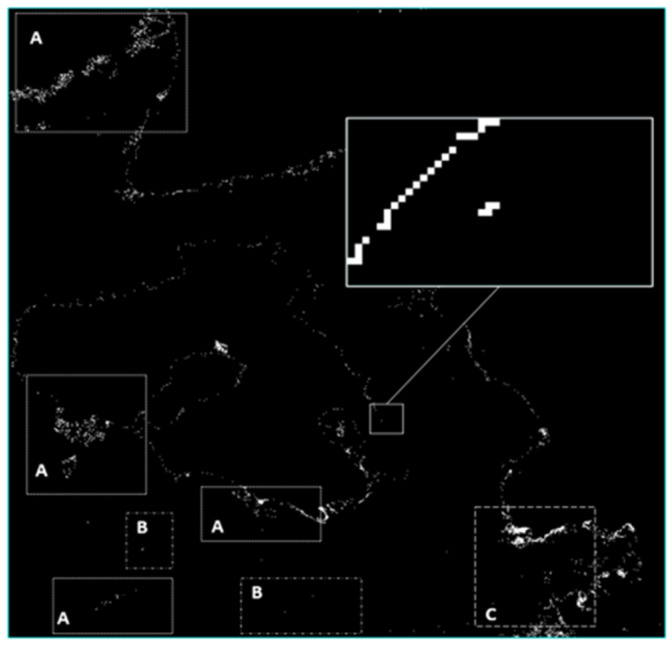
Example of a classification error on Sentinel-2 image (18 April 2019). Misidentified “plastic targets” are shown in the right middle part of the image with a bold white square. Areas labeled “A” represent commission errors due to the presence of clouds and shadows; “B” represents false detection of ships, tracks, and exhaust fumes; and “C” represents a false detection of pixels filled with intense sunshine. Almost the entire coastline is misclassified due to the effect of reflection from the bottom [44].

**Figure 4 sensors-24-05089-f004:**
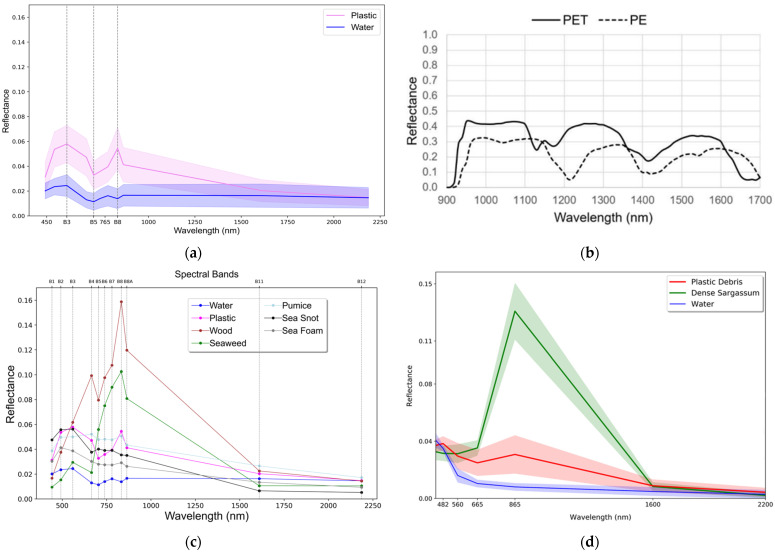
(**a**) Comparison of the spectral reflectance of all pixels containing an image of plastic objects and all pixels capturing water objects [39]; (**b**) Spectral curves of the two types of plastic [52]; (**c**) Average spectrums calculated over all pixels identified in the study [39]; (**d**) Spectrums of plastic, seaweed and water [32].

**Figure 5 sensors-24-05089-f005:**
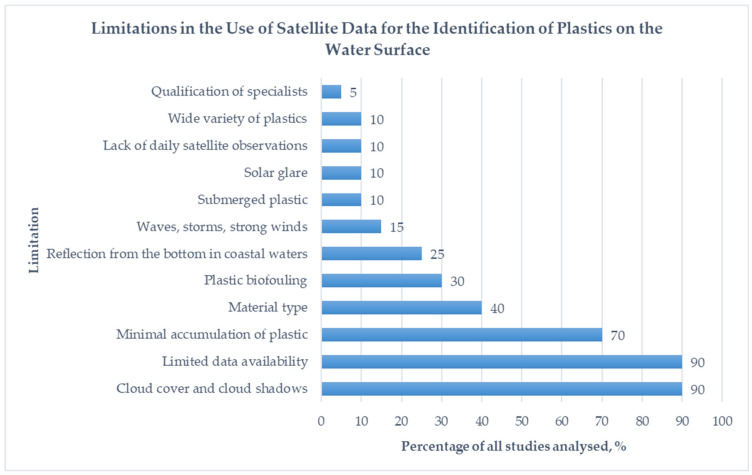
Percentage of all limitations in plastic identification from satellite imagery referred to in the submitted studies [compiled by the authors].

**Table 1 sensors-24-05089-t001:** Advantages and disadvantages of using satellite and UAV imagery. [compiled by the authors].

Technique	Advantages	Disadvantages
Satellite imagery	Large area coverage	Lower spatial resolution compared to UAVs (10–30 m from free services, 3–5 m from paid services, 30 cm on the commercial level)
	Access to a large archive of the dataset for different time periods (it is possible to view images for the past several years) [44]	Photography for a specific area is not conducted on a daily basis (usually once every 5–7 days)
	Free access to images to any territory at any time	Dependence on weather conditions: in cloudy weather, the areas to be filmed may be blocked by clouds
	The ability to survey the entire globe on a day-by-day basis	The cost is higher than with UAVs [44]
UAV imagery	Higher spatial resolution compared to satellite imagery (image accuracy can be up to 1–2 cm per pixel) [45]	Less coverage of the territory compared to satellite images
	Possibility to obtain up-to-date data on any day	Shorter flight distance (battery life and range limit the area that can be captured in a single flight)
	Convenient remote control	Geographically restricted use of UAVs (currently, UAVs are not allowed to fly over urban areas due to privacy concerns)
	The flight is below the clouds, so there is less dependence on weather conditions to take images	A well-trained person is required to launch and operate the UAV

**Table 2 sensors-24-05089-t002:** Projects in the detection of plastic in water from 2015 to 2024 [compiled by the authors].

Project Title	Project Author	Year	Method	Major Contribution
OCEAN CLEANUP [18]	Boyan Slat, Netherlands	2015	Marine expedition	The plastic is mostly in the first few meters of water
		2015	Marine expedition	More than 1.2 million plastic samples were collected from the ocean’s surface
		2016	Air expedition	Surveyed 311 km^2^ using modern sensors and RGB camera on airplane
Plastic Litter Project (PLP) [42,43]	Topouzelis, Marine Remote Sensing Group (MRSG), Mytilene, Greece	2018	Plastic targets 10 × 10 m + aerial photography from UAV + Sentinel-2 satellite imagery	Plastic can be detected on the surface of water using Sentinel-2 satellite images and UAV
		2019	Plastic targets (1 × 5 и 5 × 5 m) + aerial photography from UAV + Sentinel-2 satellite imagery	Marine debris can be detected if at least 25% of the Sentinel-2 pixel is covered with plastic
		2020	Homogeneous plastic target 10 × 10 m [46]	Large plastic targets for PLP 2021 were made
		2021	Homogeneous plastic target 10 × 10 m + aerial photography from UAV + Sentinel-2 satellite imagery [47]	An image of a clean Sentinel-2 pixel was obtained, which is completely covered by the synthetic target and contains no “empty” seawater fraction
MARIDA [48]	Kikaki, National Technical University of Athens, Greece	2021	Sentinel-2 satellite imagery and “in situ” observational data	An open-source reference dataset consisting of polygons/pixels geo-referenced to Sentinel-2 satellite imagery was developed
REMEDIES [49]	There is no information about the authors	2022	AI-enabled drones, underwater drones, fluorescent dyes, and automated microscopic detection and development and use of a marine litter monitoring app and a dashboard.	Development of a REMEDIES interoperable data repository and management portal, creating online plastic pollution and removal map for every demo site integrated into the portal.
River Cleaning Plastic [50]	Andrea and Alex Citton	2021	The river cleaning system is made up of a series of floating devices positioned diagonally on the course of the river; thus positioned, they intercept plastic waste and transport it to the river bank in a special storage area.	PILOT TEST. Over 100 kg of waste has been recovered, including various-sized plastic bottles, trays and packaging, caps, and others. The system provided crucial data and results to understand the extent of pollution, even in small watercourses.
		2022		ROSÀ PROJECT. The new system was upgraded with a surface anchoring structure. Throughout its operational period, it has been updated with the latest improvements, including the generation of electrical energy by harnessing the natural water current.
		2023		VELA 01 PROJECT. The Vela 01 system is the first to feature the new buoy modules with stainless steel shells and a grid to prevent the accumulation of algal material.
		2024		REMEDIES PROJECT. The project aims to combine technological solutions to combat pollution in river, maritime, and port areas with monitoring actions and citizen and institutional engagement on multiple levels.

**Table 3 sensors-24-05089-t003:** Central wavelength value for Sentinel-2A and Sentinel-2B satellite multispectral instrument (MSI) bands [39].

MSI Bands	Description	Sentinel-2A Central Wavelength (nm)	Sentinel-2B Central Wavelength (nm)	Spatial Resolution (m)
Band 1	Coastal Aerosol	442.7	442.3	60
Band 2	Blue	492.4	492.1	10
Band 3	Green	559.8	559	10
Band 4	Red	664.6	665	10
Band 5	Red Edge 1	704.1	703.8	20
Band 6	Red Edge 2	740.5	739.1	20
Band 7	Red Edge 3	782.8	779.7	20
Band 8	Near Infrared (NIR)	832.8	833	10
Band 8A	Narrow NIR	864.7	864	20
Band 9	Water Vapour	945.1	943.2	60
Band 10	Short Wave Infrared (SWIR) Cirrus	1373.5	1376.9	60
Band 11	SWIR 1	1613.7	1610.4	20
Band 12	SWIR 2	2202.4	2185.7	20

**Table 4 sensors-24-05089-t004:** Indexes and formulas used to investigate the possibility of detecting plastic in water.

Index	Decryption	Equation	Equation for Sentinel-2
FDI	Floating Debris Index	RE2 + (SWIR1 − RE2) × ((λ_NIR_ − λ_RED_)/(λ_SWIR1_ − λ_RED_)) × 10	B8 − (B6 + (B11 − B6)) × ((λ_B8_ − λ_B4_)/(λ_B11_ − λ_B4_)) × 10
PI [37]	Plastic Index	NIR/(NIR + Red)	B8/(B8 + B4)
NDVI	Normalized difference vegetation index	(NIR − Red)/(NIR + Red)	(B8 − B4)/(B8 + B4)
AWEI	Automated Water Extraction Index	4 × (Green − SWIR 2) − (0.25 × NIR + 2.75 × SWIR 1)	4 × (B3 − B12) − (0.25 × B8 + 2.75 × B11)
NDSI	Normalized difference snow index	(Green − SWIR1)/(Green + SWIR 1)	(B3 − B11)/(B3 + B11)
MNDWI	Modified normalized difference water index	(Green − SWIR 2)/(Red + SWIR 2)	(B3 − B12)/(B4 + B12)
NDWI	Normalized Difference Water Index	(Green − NIR)/(Green + NIR)	(B3 − B8)/(B3 + B8)
OSI	Oil spill index	(Green + Red)/Blue	(B3 + B4)/B2
WRI	Water ratio index	(Green + Red)/(NIR + SWIR 2)	(B3 + B4)/(B8 + B12)
MARI	Modified anthocyanin reflectance index	(1/Green) − (1/Red Edge 1) × Red Edge 3	(1/B3) − (1/B5) × B7

**Table 5 sensors-24-05089-t005:** Research in the field of plastic detection in water using machine learning [compiled by the authors].

Name of the Study/Project, Year	Authors	ML Techniques	Unsupervised Classification Algorithms (USAs)/Supervised Classification Algorithms (SCAs)	Attributes for Learning	Classification Accuracy, %
Development of Novel Classification Algorithms for Detection of Floating Plastic Debris in Coastal Waterbodies Using Multispectral Sentinel-2 Remote Sensing Imagery, 2021 [30]	Bidroha Basu Srikanta Sannigrahi Arunima Sarkar Basu Francesco Pilla	Support vector regression (SVR)	SCA	Sentinel-2 spectral bands: B2, B3, B4, B6, B8, B11 Indexes: NDWI, FDI	98.4
Development of automated marine floating plastic detection system using Sentinel-2 imagery and machine learning models, 2022 [53]	Srikanta Sannigrahi Bidroha Basu Arunima Sarkar Basu Francesco Pilla	Random Forest (RF)	SCA	Indexes: FDI, PI, NDVI, kNDVI Sentinel-2 spectral bands: B6, B8, B11	92–98
		Support Vector Machine (SVM)			89–100
Automatic Detection and Identification of Floating Marine Debris Using Multispectral Satellite Imagery, 2023 [39]	Miguel M. Duarte Leonardo Azevedo	XGBoost	SCA	Sentinel-2 spectral bands: B1, B8A Indexes: NDSI, MNDWI, NDWI, OSI, FDI, WRI, MARI	>95
Finding Plastic Patches in Coastal Waters using Optical Satellite Data, 2020 [38]	Lauren Biermann Daniel Clewley Victor Martinez-Vicente KonstantinosTopouzelis	Naïve Bayes	SCA	Spectral characteristics: seawater, plastic, sea foam, wood Indexes: NDWI, FDI	86
Development of Novel Classification Algorithms for Detection of Floating Plastic Debris in Coastal Waterbodies Using Multispectral Sentinel-2 Remote Sensing Imagery, 2021 [30]	Bidroha Basu Srikanta Sannigrahi Arunima Sarkar Basu Francesco Pilla	Fuzzy c-means (FCMs)	UCA	Sentinel-2 spectral bands: B2, B3, B4, B6, B8, B11 Indexes: NDWI, FDI	81.4
		K-means			82.2
Large-area automatic detection of shoreline stranded marine debris using deep learning, 2023 [34]	W. Ross Winans Yi Qiang Qi Chen Erik C. Franklin	SSD with MobileNetV2 (SS-MN)	SCA	Images of debris on the beach	71.8
Development of Novel Classification Algorithms for Detection of Floating Plastic Debris in Coastal Waterbodies Using Multispectral Sentinel-2 Remote Sensing Imagery, 2021 [30]	Bidroha Basu Srikanta Sannigrahi Arunima Sarkar Basu Francesco Pilla	Shape fuzzy c-means (SFCMs)	SCA	Sentinel-2 spectral bands: B2, B3, B4, B6, B8, B11 Indexes: NDWI, FDI	64.3
Large-area automatic detection of shoreline stranded marine debris using deep learning, 2023 [34]	W. Ross Winans Yi Qiang Qi Chen Erik C. Franklin	EfficientDet-D1 (ED-D1)	SCA	Images of debris on the beach	57.8
		Faster R-CNN	SCA		55.1

## Data Availability

No new data were created or analyzed in this study. Data sharing is not applicable to this article.

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
