# Peer review of "Review of Methods for Automatic Plastic Detection in Water Areas Using Satellite Images and Machine Learning"

_sensors, 2024, doi:10.3390/s24165089_

Round 1

Reviewer 1 Report

Comments and Suggestions for Authors

Reviewer 2 Report

Comments and Suggestions for Authors

SUMMARY:

========

Review article. Is appropriate the usual scheme of a research paper?

Has sense the section "materials and methods"? Can this be renamed?

"State of the art and outstanding methods?

It should be pointed out that Sentinel satellites do not have coverage over the oceans.

Coverage is limited to the near shore water and to inland seas (like Mediterranean).

GENERAL COMMENTS:

=================

Lines are not numbered.

Minor english corrections: "it should be noted that it’s cities

that are the main source of waste generation".

Las paragraph in section 4 seems very locally oriented.

Perhaps it can be reformulated with a more global view.

Can you insert a table in the conclusions section where "better" methods (over 90% accuracy)

are ranked (listed by decreasing accuracy), showing the most relevant parameters:

number of reference, ML technique and features used ("attributes for learning", bands, indexes...).

DETAILS:

========

Table 1. Correct units at second line (size of targets).

Section 2.3. Define NDVI index.

Figure 3.a ==> Are lambda values for the vertical blue lines correct??

What is the symbol between: "𝑅𝑅𝐸2" and "𝑅𝑆𝑊𝐼𝑅1" (и), in section 2.5?

Is it possible to create a table with equations defining the indexes

mentioned in section 2.5 (NDSI, MNDWI, NDWI, OSI, FDI, WRI, MARI)?

Comments on the Quality of English Language

Minor english corrections: "it should be noted that it’s cities that are the main source of waste generation".

Reviewer 3 Report

Comments and Suggestions for Authors

The paper primarily focuses on summarizing research that utilizes satellite imagery to detect plastic materials in the ocean. For a review paper, it would be beneficial to provide a high-level summary of the various approaches used for detecting plastics, including physically-based indices and machine learning techniques. The paper should also discuss the respective advantages and disadvantages of these methodologies.

Additionally, while the paper currently only addresses studies using multispectral images, it should also consider those employing other data types, such as SAR or microwave sensing.

Comments on the Quality of English Language

Many minor grammatical errors are observed. Please proofread the paper carefully and update accordingly.

Round 2

Reviewer 1 Report

Comments and Suggestions for Authors

The authors have basically addressed my concerns. My final suggestion is to provide an overall framework figure in the main body of the paper, as I pointed out in my first comment. Providing a graphical abstract is certainly beneficial for the article, but the overall framework figure in the main body of the paper is even more indispensable. Unlike graphic abstract, it should contain more textual information.

Author Response

We have considered your comment. An overview scheme representing the sequential steps of the plastic detection technology was presented in Section 2.6. We have improved this scheme and presented it at the end of the introduction. Relevant changes are highlighted in blue. Thank you very much! 

Reviewer 3 Report

Comments and Suggestions for Authors

Thank you for the modification. The paper now covers more comprehensive scope of plastic detection using different types of sensors and platforms.

Author Response

Thank you very much! The last change we made to the manuscript is that an overview diagram with the sequential steps of the technology we described is now presented at the end of the introduction. Relevant changes are highlighted in blue.